# Stimulation of Fibronectin Matrix Assembly by Lysine Acetylation

**DOI:** 10.3390/cells9030655

**Published:** 2020-03-08

**Authors:** Maria E. Vega, Birgit Kastberger, Bernhard Wehrle-Haller, Jean E. Schwarzbauer

**Affiliations:** 1Department of Molecular Biology, Princeton University, Princeton, NJ 08544, USA; mevega@princeton.edu; 2Department of Cell Physiology and Metabolism, Centre Médical Universitaire, 1 Rue Michel-Servet, CMU, 1211 Geneva 4, Switzerland; Birgit.Kastberger@outlook.com (B.K.); Bernhard.Wehrle-Haller@unige.ch (B.W.-H.)

**Keywords:** diabetic nephropathy, extracellular matrix, fibrosis, fibronectin, β1 integrin, kindlin-2

## Abstract

Diabetic nephropathy, a devastating consequence of diabetes mellitus, is characterized by the accumulation of extracellular matrix (ECM) that disrupts the kidney’s filtration apparatus. Elevated glucose levels increase the deposition of a fibronectin (FN) matrix by mesangial cells, the primary matrix-producing cells of the kidney, and also increase acetyl-CoA leading to higher levels of lysine acetylation. Here, we investigated the connection between acetylation and the ECM and show that treatment of mesangial cells with deacetylase inhibitors increases both acetylation and FN matrix assembly compared to untreated cells. The matrix effects were linked to lysine 794 (K794) in the β1 integrin cytoplasmic domain based on studies of cells expressing acetylated (K794Q) and non-acetylated (K794R) mimetics. β1(K794Q) cells assembled significantly more FN matrix than wildtype β1 cells, while the non-acetylated β1(K794R) form was inactive. We show that mutation of K794 affects FN assembly by stimulating integrin-FN binding activity and cell contractility. Wildtype and β1(K794Q) cells but not β1(K794R) cells further increased their FN matrix when stimulated with deacetylase inhibitors indicating that increased acetylation on other proteins is required for maximum FN assembly. Thus, lysine acetylation provides a mechanism for glucose-induced fibrosis by up-regulation of FN matrix assembly.

## 1. Introduction

Diabetes is a chronic disease that causes progressive damage to organs and tissues. One of the major complications of diabetes is diabetic nephropathy, which is the leading cause of end-stage renal disease in the United States accounting for about 40% of newly diagnosed cases [1]. The key histological feature of diabetic nephropathy is fibrosis, the excess deposition of ECM particularly in the glomeruli where mesangial cells are the primary matrix-producing cells [2]. The mesangial ECM organizes and supports the glomerular capillary network, and ECM expansion occludes spaces required for blood filtration culminating in non-functional glomeruli. Major components of the mesangial ECM are fibronectin (FN) and collagens. Excess amounts of these proteins are found in renal biopsies of patients with this disease [3], and cultured mesangial cells exposed to elevated glucose levels show a significant up-regulation of FN matrix assembly [4]. We have previously shown that high glucose levels increase FN matrix by activating mesangial cell integrin receptors through a protein kinase C-dependent pathway [4].

FN is an abundant and ubiquitous ECM protein important for interactions between cells and the surrounding matrix. FN matrix is assembled by a stepwise process that is initiated by α5β1 integrin receptors binding to FN dimers, which induces FN conformational changes and promotes FN–FN interactions to form nascent fibrils [5]. Through continued FN deposition, these fibrils grow and mature into a stable, insoluble matrix. Importantly, the FN matrix acts as a framework for the deposition of collagens and other key ECM proteins [6,7,8,9]. Therefore, deregulation of FN matrix assembly could have devastating effects on the overall organization, abundance, and composition of ECM fibrils causing a fibrotic response [10,11]. The slow turnover of the ECM would further exacerbate the deleterious effects on the glomerular filtration apparatus [12].

Elevated glucose conditions increase glycolysis and the production of acetyl-CoA, which then serves as a substrate in protein acetylation reactions [13]. Protein acetylation is an important post-translational modification of lysine that can alter the functional profile of proteins by influencing catalytic activity, capacity to interact with other molecules, subcellular localization, and/or stability. This reversible modification is regulated by acetyltransferases and deacetylases and can change protein functions in diverse ways by neutralizing the positive charge on lysine and affecting protein interactions or conformation [14,15]. High levels of acetylation on lysines are observed in mouse diabetic kidney protein extracts and in mesangial cells grown under high glucose conditions [16,17,18]. Treatments that activate deacetylases thus decreasing lysine acetylation in a diabetic rat model lowered renal fibrosis compared to control animals [19,20] suggesting that acetylation can induce a fibrotic response. Acetylation has been shown to indirectly affect FN matrix in submandibular glands through the effects of microtubule acetylation on integrin adhesion complexes [21]. Changes in acetylation could have direct effects on FN matrix formation by changing the functions of critical matrix assembly proteins. High resolution mass spectrometry analysis identified lysine acetylation sites in many proteins including the cytoplasmic domain of β1 integrin [22]. Here, we investigate the effects of increased lysine acetylation on FN assembly by mesangial cells and identify a possible role for β1 integrin modification in cell-FN interactions and downstream FN matrix assembly.

## 2. Materials and Methods

### 2.1. Fibronectin, Inhibitors, and Antibodies

Human fibronectin was purified from fresh-frozen human plasma by gelatin-sepharose affinity chromatography and dialyzed into CAPS buffer [23]. Cells were treated with either suberoylanilide hydroxamic acid (SAHA) or suberohydroxamic acid (SBHA) from a 10 mM stock in DMSO (Sigma-Aldrich, St. Louis, MO, USA). Analysis of acetyl-tubulin levels at increasing doses of inhibitor showed that a final concentration of 5 μM was effective at inhibiting deacetylation while having no cytotoxic effects. The following primary antibodies were used: rabbit polyclonal anti-FN (R184) antiserum raised in-house against recombinant FN repeats III1-6, mouse monoclonal anti-human FN concentrated supernatant (HFN7.1, DSHB, Iowa, USA), purified rabbit monoclonal anti-β1 integrin (D6S1W, Cell Signaling, Danvers, MA, USA), purified mouse monoclonal anti-acetyl-lysine antibody (AAC03 and AAC03-HRP, Cytoskeleton, Inc, Denver, CO, USA), purified mouse monoclonal anti-acetyl-tubulin (clone 6-11B-1, Sigma-Aldrich), purified mouse monoclonal kindlin-2 (Clone 3A3, Sigma-Aldrich), mouse monoclonal anti-α-tubulin ascites fluid (Clone B-5-1-2, Sigma-Aldrich), and purified rabbit polyclonal anti-GAPDH (14C10, Cell Signaling). Secondary antibodies used were horseradish peroxidase-conjugated (HRP) goat anti-rabbit IgG, HRP-goat anti-mouse IgG (both from ThermoFisher Scientific, Waltham, MA, USA) and Alexa-Fluor-488- and 568-conjugated goat anti-mouse IgG or goat anti-rabbit IgG (Invitrogen). For actin filament detection, Texas Red-X Phalloidin was used (Invitrogen, Carlsbad, CA, USA).

### 2.2. Mesangial Cell Culture

Conditionally immortalized mesangial cells [24] were maintained at 33 °C in DMEM medium containing 10% fetal bovine serum (FBS, Hyclone, Logan, UT, USA), 100 IU/mL interferon-γ, 20 mM glucose, 10 mM mannitol, 1 mM sodium pyruvate, 100 U/mL penicillin, 100 μg/mL streptomycin, and 0.25 μg/mL amphotericin B. Differentiation to a phenotype similar to freshly isolated primary mesangial cells was induced by altering the culture temperature to 37 °C and removing interferon-γ [25]. After 4 d at 37 °C, cells were passaged and allowed to attach for 4 h before serum starving for 20 h in media containing 5 mM glucose with 25 mM mannitol. After starvation, cells were conditioned for 24 h in DMEM with 10% FBS and 5 mM glucose plus 25 mM mannitol or 30 mM glucose for 24 h and then re-plated for experiments.

### 2.3. Generation of Cells Expressing β1 Integrin Lysine Mutations

The cDNA encoding human β1 integrin was obtained from RZPD (Deutsches Ressourcenzentrum fur Genomforschung GmbH, Berlin, Germany) Clone ID: IRATp970E0719D6. It was cloned into pCDNA3 (Invitrogen) using XbaI and EcoRI sites. Mutations were introduced by primer overlap extension and subsequently verified by automated sequencing. The extracellularly tagged β1 integrin was constructed by introducing PinAI and XhoI sites in a loop in the hybrid domain by duplicating proline at position 88 (AEGKP^88^EDIT), which resulted in the AEGK**P*VSRG*P**EDIT sequence that was used to insert the EGFP sequence (forward:5-AACCGGTCTCTCGAGGACCAGAGGATATTACTCA-3; reverse: 5-TCCTCGAGAGACCGGTTTGAGCTTCTCTGCTGTTC-3). For stable transfections of GD25 cells, and in order to avoid an excessive, but maintaining a moderate, long-term transcription in stably transfected cells, the CMV promoter of the pcDNA3 vector was replaced by a 1kb fragment of the matrix attachment region of chicken lysozyme (MAR, kind gift of N. Mermod, Lausanne, Switzerland) [26,27], and a 1.4 kb fragment (NheI-EcoRI) of the human β-actin promoter. Since GD25 cells are resistant to G418, the neomycin resistance gene in the modified β1 integrin containing pcDNA3 (sm-pcDNA3) was swapped for that of the puromycin gene (smp-pcDNA3). To evaluate the effects of β1 integrin lysine acetylation, two recombinant mutants were produced in which lysine residues were replaced either with glutamine (Q) to mimic acetylation or arginine (R) to mimic non-acetylation of lysine K794. DNA sequence analysis was performed for all constructs to ensure error-free amplification and correct base replacement.

Mouse β1 integrin deficient GD25 cells [28] (a kind gift of Dr. R. Faessler, Munich, Germany) were grown in DMEM, 10% FBS, 100 U/mL penicillin, 100 μg/mL streptomycin, and 0.25 μg/mL amphotericin B. Transfections were performed with Jet PeI (Polyplus Transfection, New York, NY, USA) according to the manufacturer’s recommendation and stably transfected GD25 cells were selected in medium supplemented with 4 μg/mL puromycin. Transfectants were subsequently sorted by FACS for the surface expression of the human β1 integrin. Cells were detached with trypsin/EDTA, blocked in culture medium, and incubated with mouse anti-human β1 integrin antibody (Ts2/16; Biolegend, San Diego, CA, USA) for 45 min on ice. Cells were washed, resuspended in DMEM and incubated with goat anti-mouse PE (Phycoerythrin)-conjugated secondary antibody for 30 min on ice. Cells were washed twice and resuspended in PBS for the FACS, which was performed with a BD FACSAria II instrument (selecting between 10,000 and 100,000 cells for reseeding). Cells were maintained in medium supplemented with 4 μg/mL puromycin. β1 integrin transgenes in sorted cell lines were confirmed by sequencing of PCR products generated from genomic DNAs. Cell surface levels of β1 integrins on stably transfected GD25 cell lines were measured by flow cytometry. Cells were treated as described for FACS using either Ts2/16 antibody (1:500) or rat anti-mouse β1-integrin 9EG7 (1:200) (BD Transduction Laboratories, San Jose, CA, USA). Fifty thousand cells were analyzed for each antibody.

### 2.4. Isolation of FN Matrix and Immunoblotting

Pre-conditioned mesangial cells were plated in medium with 20 μg/mL human plasma FN and either 5 mM glucose and 25 mM mannitol or 30 mM glucose at 7.6 × 10^5^ cells per 35 mm dish. Cells were treated with either 5 μM SAHA, SBHA, or DMSO for 24 h. For GD25 cells, 5 × 10^5^ cells were grown in medium for 24 h and then treated with 5 μM SBHA or DMSO for the next 24 h before lysis. Confluent cells were washed with cold phosphate buffered saline (PBS) and then lysed in deoxycholate (DOC) lysis buffer (2% DOC, 20mM Tris-HCl, pH 8.8, 2 mM EDTA, and protease inhibitor cocktail (Roche)) according to our standard procedure [29]. Equal amounts of DOC-soluble protein or proportional volumes of DOC-insoluble material were separated by SDS-PAGE on 6% (for FN) or 10% (for GAPDH and acetyl-tubulin) polyacrylamide gels. For some analyses of acetyl-tubulin, cells were lysed in RIPA buffer (50 mM Tris-HCl, pH 7.4, 1% NP-40, 0.5% Na-DOC, 0.1% SDS, 150 mM NaCl, 2 mM EDTA, plus protease inhibitor cocktail (Roche, Indianapolis, IN, USA). Protein was then transferred onto a nitrocellulose membrane and subsequent incubations were performed in buffer A (25 mM Tris-HCl, pH 7.5, 150 mM NaCl, 0.1% Tween-20). Antibodies were used at these dilutions: HFN7.1 (1:2000), R184 (1:50,000), acetyl-tubulin (1:5000), tubulin (1:50,000), and GAPDH (1:5000). Secondary antibodies were diluted 1:10,000 in buffer A. Blots were developed using SuperSignal West Pico Plus ECL Reagents (ThermoScientific). Densitometry was performed on scanned films using Adobe^®^ Photoshop^®^ Software, and exposures yielding signals within the linear range were quantified. Fibronectin and total tubulin levels were normalized to GAPDH. Fold-change and SEM were calculated from the means of three independent experiments.

### 2.5. Cell Attachment and Ligand Binding Assays

Conditioned mesangial cells in 5 mM or 30 mM media and GD25 cells were treated with 5 μM SBHA or DMSO for 24 h. Cells were then plated onto coverslips or non-tissue culture treated plastic wells coated with a 10 μg/mL solution of plasma FN overnight at 4 °C and blocked in 1% BSA/PBS for 30 min at room temperature. In total, 5.0 × 10^4^ pre-treated cells were plated in the appropriate medium without serum [4] but including 5 μM SBHA or DMSO. Cells were allowed to attach for 30 min at 37 °C, then washed and fixed with 3.7% formaldehyde in PBS, and imaged by phase contrast microscopy or stained with Texas Red-X Phalloidin for immunofluorescence imaging.

Cells were conditioned and treated as described above and then used in a ligand-binding assay as described [4]. Fifty microliter of samples were layered onto 600 μL of 40% sucrose in PBS and centrifuged at 10,000 rpm for 5 min [30]. Pellets were resuspended in 4% SDS sample buffer and boiled for 10 min. Samples were then reduced and separated by SDS-PAGE on a 6% polyacrylamide gel and immunoblotted for human FN, or 10% polyacrylamide gel for GAPDH loading control.

### 2.6. Matrix Contraction Assay

Fibrin-FN matrices were prepared as described previously [31,32] with the following final concentrations of components: 600 μg/mL human fibrinogen (BioVision Inc, Milpitas, CA, USA), 150 mM NaCl, 50 mM CaCl_2_, and 50 mM Tris-HCl, pH 7.4. Assessment of contaminants in the fibrinogen stock identified FN and Factor XIII so further addition of these components was unnecessary for clot formation. GD25 cells were mixed in at 4 × 10^6^ cells/mL followed by 2 U/mL of thrombin and the mixture was rapidly pipetted into the well of a 48-well plate previously coated with 1% bovine serum albumin (BSA, Sigma-Aldrich) in PBS. After 30 min at 37 °C, 500 μL of DMEM was added and contraction was measured as in [31].

### 2.7. Immunofluorescence Microscopy

Mesangial cells conditioned in 30 mM glucose medium were seeded on glass coverslips in medium containing 10 μg/mL human plasma FN and 5 μM SBHA or DMSO. Seeding densities were 4.8 × 10^5^ cells/12 mm coverslip for 8 h, 3.2 × 10^5^ cells/coverslip for 16 h, or 1.6 × 10^5^ cells/coverslip for 24 h of growth. GD25 cells were seeded at 2 × 10^5^ cells/coverslip for 24 h and then treated with 5 μM SBHA or DMSO for an additional 24 h. Confluent cells were fixed and stained as described [4]. All antibodies were diluted in 2% ovalbumin in PBS and incubated at 37 °C for 30 min. Primary antibodies were used at the following dilutions: HFN7.1 (1:100), R184 (1:100), and Alexa-Fluor-488-conjugated secondary antibodies (1:600). Coverslips were then mounted using ProLong Gold anti-fade reagent (Life Technologies, Grand Island, NY, USA). All images were captured using a Nikon Eclipse Ti microscope equipped with a Hamamatsu C10600 ORCA-R2 digital camera. Mean fluorescence measurements were performed using ImageJ on 6 randomly selected fields per condition. Background fluorescence was removed from images using a rolling ball radius of 50. Representative fields are shown. Fluorescence fold-change and SEM were calculated from three independent experiments. Images were adjusted equally using Adobe^®^ Photoshop^®^ software.

### 2.8. Acetyl-lysine Immunoprecipitation

GD25 β1(null) and β1(WT) cells were grown in media with 5 μM SBHA or DMSO for 24 h before lysis and immunoprecipitation using Signal-Seeker™ Acetyl-Lysine Detection Kit (Cytoskeleton, Inc, Denver, CO, USA) following the manufacturer’s instructions. The immunoprecipitation eluates were separated by 8% polyacrylamide SDS-PAGE and transferred onto a nitrocellulose membrane. Blots were probed for both β1 integrin (1:2000) and mouse monoclonal acetyl-lysine-HRP (1:3000, Cytoskeleton, Inc.) and developed as described above.

### 2.9. TIRF Microscopy

GD25 cells were plated onto the 14 mm diameter surface of a glass-bottom dish (Mat-Tek, Corp, Ashland, MA, USA) that had been coated with a 10 μg/mL solution of plasma FN overnight at 4 °C and blocked in 1% BSA/PBS for 30 min at room temperature. In total, 5.0 × 10^4^ cells were plated in complete medium for either 4 h or 24 h before fixing with 3.7% formaldehyde in PBS and permeabilizing in 0.5% Triton-X 100 in PBS for 15min. Cells were stained with anti-kindlin-2 antibody (1:100) and Alexa-Fluor-568-conjugated secondary antibody (1:600). TIRF microscopy was performed on Nikon Ti-E with Perfect Focus System (PFS) fitted with an Andor iXon Ultra CCD camera (Oxford Instruments). Images were captured using NIS-Elements Software (Nikon, Melville, NY, USA). Total kindlin-2 and β1 integrin fluorescence intensities per cell were quantified using Image J for 30 cells per condition. Corrected fluorescence intensity values for individual cells were calculated as follows: total cell fluorescence intensity – (cell area × mean fluorescence of background), and then kindlin-2 fluorescence was normalized to β1 integrin fluorescence for each cell. Kindlin-2 to β1 integrin ratios were graphed as box-plots with means and medians. Mesangial cells conditioned to 5 mM or 30 mM glucose were plated onto the 14 mm diameter surface of a glass-bottom dish (Mat-Tek Corp) coated with a 10 μg/mL solution of plasma FN overnight at 4 °C and blocked in 1% BSA/PBS for 30 min at room temperature. In total, 5.0 × 10^4^ cells were plated in complete medium for 2 h before fixing with 3.7% formaldehyde in PBS and permeabilizing in 0.5% Triton-X 100 in PBS for 15 min. Cells were stained with anti-acetyl-lysine antibody (1:500) and Alexa-Fluor-488 conjugated secondary antibody (1:600), before imaging by TIRF microscopy as described above.

### 2.10. Statistical Analysis

Results are reported as the mean ± standard error for a minimum of three independent experiments. Statistical analyses were performed using a one-way ANOVA, with *p* < 0.05 considered statistically significant.

## 3. Results

### 3.1. Effects of Deacetylase Inhibitors on FN Matrix Assembly

To confirm that elevated glucose levels cause an increase in lysine acetylation in mesangial cells, we analyzed the levels of acetylated tubulin grown under normal (5 mM) and high (30 mM) glucose concentrations. The acetyl-tubulin level was very low in normal glucose but increased dramatically when cells were cultured in high glucose (Figure 1A). Immunoblotting cell lysates with an anti-acetyl-lysine antibody identified additional acetylated proteins in mesangial cells (data not shown) and cytoplasmic staining with anti-acetyl-lysine antibodies was enhanced after cell growth in high glucose medium (Appendix A). High glucose conditions also increased the expression of total tubulin by about 2-fold (Figure 1A). Taken together, our results confirm reports of others [16,18] that growth of mesangial cells in high glucose medium stimulates lysine acetylation.

We previously showed that mesangial cells grown in high (30 mM) glucose conditions assemble significantly more FN matrix than cells in normal (5 mM) glucose [4] (Appendix A). Protein acetylation could be a contributing factor to this increase in matrix assembly. To increase acetylation independently of glucose concentration, mesangial cells were treated with various histone deacetylase (HDAC) and SIRT1 inhibitors and the effects on tubulin acetylation were compared (data not shown). The HDAC inhibitors SAHA and SBHA were the most effective. Indeed, both inhibitors dramatically increased acetyl-tubulin (Figure 1B) and total lysine acetylation (Appendix A), compared to the DMSO control cell lysates. Quantitative PCR analysis of FN mRNA levels showed that neither glucose concentration nor HDAC inhibitors affected FN expression (data not shown). To address the effects of HDAC inhibition on FN matrix assembly, we treated mesangial cells in 30 mM glucose with 5 μM SAHA or SBHA with exogenous human FN for 24 h and then FN matrix levels were quantified using a deoxycholate (DOC) solubility assay to separate nascent DOC-soluble FN fibrils from stable DOC-insoluble FN matrix. We observed a significant increase in the DOC-insoluble FN matrix from cells treated with SAHA (2.8 ± 0.4-fold compared to DMSO) or with SBHA (3.1 ± 0.3-fold compared to DMSO) (Figure 1C), indicating that increased acetylation promotes formation of insoluble FN matrix. Immunofluorescence analyses showed differences in matrix formation at times even earlier than 24 h. SBHA-induced differences in the number of FN fibrils were detected at 8 and 16 h of treatment (1.37-fold and 1.20-fold higher, respectively, versus DMSO) (Figure 1D). The changes in FN matrix at early time-points suggest that acetylation is affecting the initial stages of FN assembly, which relies on integrin-dependent interactions.

### 3.2. Mimetics of Acetylated and Non-Acetylated Lysine in β1 Integrin

Previous studies have identified an acetylation site in the β1 integrin cytoplasmic tail at lysine 794 (K794) in the membrane-distal NPxY motif (Figure 2A) [22]. To test whether modification of K794 alters FN assembly, we generated two mutations at this site in the β1 integrin: lysine to glutamine (K794Q) to mimic an acetylated lysine and lysine to arginine (K794R) to prevent acetylation but retain a positively charged residue at that location [22]. In order to localize the transfected integrins, a GFP tag was also inserted into the extracellular domain of the integrin [33]. Wild type and mutant integrin constructs were transfected into a β1 integrin-null mouse fibroblast cell line (GD25) and stably transfected cells were isolated. We confirmed the expression of wild type and mutant integrins by immunoblotting cell lysates with a β1 integrin antibody (Appendix A) and by flow cytometry for total integrins (with Ts2/16 antibody) and activated integrins (with 9EG7 antibody) (Appendix A). Total integrin expression and cell surface levels of integrins were comparable among the three β1 transfected cell lines. Additionally, wild type and mutant integrin localization to focal adhesions was confirmed by TIRF microscopy of the GFP integrins in cells plated onto FN-coated coverslips (Appendix A). Parental GD25 null cells showed no GFP signal. No significant difference in the amount of endogenous FN secreted by these cells was noted (data not shown).

Cells expressing wild type or mutant integrins were grown to confluence for 2 days and then DOC-insoluble FN matrix was isolated and analyzed. As expected, the expression of integrin β1(WT) restores FN matrix assembly to the null cells (Figure 2B). Cells expressing the acetyl-mimic β1(K794Q) integrin had the highest amount of FN matrix compared to the β1(WT) cells (2.8 ± 0.7-fold). β1(K794R) cells showed negligible levels of DOC-insoluble FN, similar to the null cells (Figure 2B). Immunofluorescence staining of the FN matrix assembled by confluent cells (Appendix A) corroborated the DOC solubility assay data. Fibrillar FN matrix was assembled by β1(K794Q) cells and β1(WT) cells, albeit at different levels (Figure 2C), while only small non-fibrillar aggregates of FN were detected in null and β1(K794R) cell cultures (Appendix A). Higher matrix levels with β1(K794Q) cells indicate that acetylation enhances FN assembly whereas the reduced amount of matrix assembly by β1(K794R) cells indicates that the inability to be modified negatively affects the ability of integrins to assemble FN. Taken together, our data implicate integrin acetylation in glucose-mediated increases in FN matrix deposition.

### 3.3. β1 Integrin Acetylation in SBHA-Treated Cells

To determine whether SBHA treatment of cells stimulates β1 integrin acetylation, we used an acetyl-lysine immunoprecipitation protocol with GD25 β1(WT) and GD25 β1(null) cells, treated or not with SBHA. We observed an enrichment of acetylated proteins (in particular, acetyl-tubulin but also other acetyl-lysine-positive proteins) in lysates of cells treated with SBHA compared to control treated cells (Figure 3A). Immunoblotting of the acetyl-lysine immunoprecipitates with anti-β1 integrin antibodies showed a prominent band in the β1(WT)/SBHA sample compared to a faint band without SBHA treatment (Figure 3B). While no β1 integrin band was detected in the acetyl-lysine immunoprecipitate of the null cells in either DMSO or SBHA condition, the presence of a faint band in the β1(WT)/DMSO lane suggests that there is acetylated β1 integrin in cells without deacetylase inhibitor treatment. The background band at ~90 kD was only observed in the null cell lysates and was independent of SBHA treatment. Taken together, these results demonstrate increased acetylation of β1 integrin in cells treated with an HDAC inhibitor.

### 3.4. Effects of Increased Acetylation on Cell Attachment to FN

The differences in FN matrix produced by cells treated or not with SBHA and cells expressing the mutant integrins suggest that modification at K794 modulates the ability of integrins to bind to FN. This idea was tested by a cell attachment assay. Mesangial cells were grown in either 5 mM or 30 mM glucose and treated with SBHA before trypsinization, and re-plating onto FN-coated surfaces was done. Cells were allowed to attach for 30 min, unattached cells were removed, and attached cells were fixed and counted by phase contrast microscopy (Figure 4A,B). These data confirm our previous findings [4] showing almost 2-fold more attachment of mesangial cells grown in high glucose compared to those grown in normal glucose (1.8 ± 0.3-fold). Interestingly, SBHA-treated cells showed significantly more attachment than untreated counterparts, independent of the glucose concentration −2.5 ± 0.5-fold in 5 mM and 1.6 ± 0.2-fold in 30 mM glucose media. GD25 cell β1 mutants also showed differences in cell attachment that correlated with acetylation. Compared to β1(WT) cells, the cells expressing β1(K794Q) acetyl-mimic had almost 2-fold more attachment (1.9 ± 0.1-fold) while attachment was reduced by half for non-acetylated β1(K794R) cells (0.5 ± 0.1-fold) (Figure 4C). These attachment data from two distinct cell types support the possibility that β1 integrin acetylation state affects the ability of cells to attach to a FN-coated substrate.

To determine the potential effects of acetylation on cell spreading, GD25 β1 mutant cells were grown on a FN-coated surface for 4 h and actin stress fibers were detected by phalloidin staining. We observed comparable actin stress fibers and cell spreading between β1(WT) and β1(K794Q) cells. β1(K794R) cells, however, were less spread with no observable stress fibers compared to β1(WT) cells at this time point (Figure 4D). Analyses at later times showed that β1(K794R) cell spreading was delayed by 2–3 h compared to the other cells and eventually all cell lines were spread equivalently (data not shown). These results indicate that, along with integrin-mediated attachment to FN, mutation at K794 also affects actin stress fiber formation. Since matrix assembly depends on integrin complexes connecting actin filaments to FN fibrils, our findings raise the possibility that acetylation at K794 might promote FN fibril assembly by modulating integrin-FN binding strength through the recruitment of cytoskeleton-associated molecules.

### 3.5. Differential Recruitment of Kindlin-2 to Focal Adhesions

FN matrix assembly depends on polymerization of FNs bound to α5β1 integrins and polymerization of actin connected to integrin cytoplasmic tails via focal adhesion adapter proteins. β1 integrin contains two NPxY motifs for adapter protein binding, a membrane-proximal NPIY motif within the site for talin binding and membrane-distal NPKY containing K794 that is associated with binding to kindlins [34]. We analyzed kindlin-2 localization with integrins in GD25 β1 mutant cells. Kindlin-2-positive focal adhesions were detected in most of the β1(K794Q) and β1(WT) cells, but in only some β1(K794R) cells by epifluorescence microscopy (Figure 5A, Appendix A). TIRF microscopy was used to analyze the localization of β1 integrin via its GFP tag relative to kindlin-2 immunostaining in the β1(K794Q) and β1(K794R) cells. All GFP-positive focal adhesions formed by cells expressing the acetyl-mimic β1(K794Q) also contained kindlin-2 (Figure 5B, Appendix A). Interestingly, in TIRF microscopy of β1(K794R) cells, the kindlin-2 signal was fainter in integrin-positive focal adhesions and, in some cases, it was not present (Figure 5B, Appendix A). Differences in localization are not due to expression changes since kindlin-2 levels were equivalent by immunoblot in all β1 integrin-transfected cells (Appendix A). Quantification of the kindlin-2 fluorescence intensity normalized to the integrin GFP signal showed significantly less localization of kindlin-2 at focal adhesion sites in β1(K794R) compared to β1(K794Q) cells (Figure 5C). Taken together, these data suggest that β1 integrin modification at the K794 site is important in promoting recruitment of kindlin-2.

### 3.6. Effects of Acetylation on Integrin Activity and Cell Contractility

Cell attachment and focal adhesion results suggest that deacetylase inhibition affects integrin activity. Integrin affinities for FN were compared by measuring FN binding to cells in suspension [35]. Soluble FN was incubated with mesangial cells that had been grown in either 5 mM or 30 mM glucose with or without SBHA. Analysis of bound FN by immunoblotting shows that the SBHA treatment significantly increased FN binding capacity in both growth conditions compared to control treatments (Figure 6A,C). As previously reported [4], we observed increased FN binding in 30 mM/DMSO compared to 5 mM/DMSO conditions by about 4-fold. Similar results were obtained by comparison of FN bound to β1(K794Q) cells with β1(WT) cells (Figure 6B,C). β1(K794R) cells bound much less FN than β1(WT). As expected, the null cells did not bind FN, confirming the requirement for β1 integrin for FN binding. Higher FN binding to mesangial cells with increased acetylation and to GD25 cells expressing β1(K794Q) integrin indicate that integrin activity is enhanced by conditions that promote increased lysine acetylation.

Another measure of integrin-FN binding is a cell contractility assay. Cell-dependent contraction of a fibrin-FN matrix was used to compare β1(WT), β1(K794R), and β1(K794Q) cells. The results resemble the behavior observed in FN binding, cell attachment, and matrix assembly experiments. β1(K794Q) cells showed the highest contraction, followed by β1(WT) and then β1(K794R) cells (Figure 6D). Therefore, by a number of different metrics, we show that K794Q mutation promotes integrin activation in order to enhance FN matrix assembly.

### 3.7. Stimulation of Lysine Acetylation Enhances the Effects of β1(K794Q) on FN Matrix Assembly

Many intracellular proteins can be acetylated, as illustrated by the broad spectrum of protein bands detected by anti-acetyl-lysine antibodies after SBHA treatment (Appendix A). To determine if acetylation of proteins other than β1 integrin contributes to matrix assembly, GD25 cells expressing mutant integrins were treated with SBHA or not. The effectiveness of the SBHA treatment was illustrated by immunoblotting for acetyl-tubulin, which showed a dramatic increase in all cell lines with SBHA (Figure 7A). DOC-insoluble samples were immunoblotted to detect FN matrix and showed a significant increase in insoluble FN by SBHA for both β1(WT) and β1(K794Q) cell samples (Figure 7A). FN matrix assembled by β1(K794Q) cells was at least 3-fold higher than β1(WT) FN matrix both with and without SBHA treatment. Mean fluorescence intensities of FN matrix in SBHA-treated β1(WT) and β1(K794Q) cell cultures also increased (Figure 7B) indicating that acetylation of lysines other than the K794 site further stimulates matrix assembly. Treatment of the β1(K794R) cells with SBHA had no effect on DOC-insoluble FN levels (Figure 7A,B), showing that stimulation of FN assembly depends on the state of K794 in the β1 integrin tail. Taken together, these data show that while β1(K794Q) integrin supports FN matrix assembly, this modification is not sufficient in that treatments that promote acetylation of other proteins further increase FN matrix assembly.

## 4. Discussion

Increases in ECM deposition and lysine acetylation occur in the kidney when glucose levels are high. In this study, we investigated the connection between these two processes and found that lysine acetylation stimulates FN matrix assembly. Treatment of mesangial cells grown in either normal or high glucose conditions with deacetylase inhibitors significantly increased cell attachment to FN and FN matrix levels. Similarly, GD25 cells expressing an integrin acetylation-mimetic mutation at K794 (β1(K794Q)) showed higher cell attachment and FN matrix assembly compared to cells with wild type or non-acetylated β1 mutants. Measurements of integrin-FN binding and cell contractility demonstrated that acetylation enhances integrin activity. In contrast, the inability to acetylate β1 in cells expressing the β1(K794R) mutant may explain the critically hampered assembly of FN matrix, even with HDAC inhibitor treatment. Apparently, acetylation on at least some of the integrins is necessary for matrix assembly. Expression of β1(WT) or β1(K794Q) was not sufficient for maximum matrix assembly since HDAC inhibitor treatment further stimulated FN matrix assembly by these cells, indicating that modification of other proteins contributes to the FN matrix assembly process. Taken together, our results demonstrate that lysine acetylation has an important role in increasing FN matrix assembly. Combined with our previous work on high glucose-induced ECM accumulation, we propose that multiple pathways downstream of glucose act to increase integrin-mediated FN matrix assembly and suggest that FN assembly defines a common final pathway in the development of kidney fibrosis.

FN fibrils form the foundation for the normal deposition of other ECM proteins including collagens [5,6,7,8,9]. In fibrosis, however, ECM protein levels can change and assembly is deregulated resulting in excess deposition of ECM proteins and accumulation of disorganized networks of fibrils of FN and collagens [11,12,36] as shown in glomerulosclerosis [4,10,37]. Since FN assembly depends on integrins, primarily α5β1 integrin, intracellular pathways that respond to glucose and act on integrins are obvious candidates to go awry and perturb matrix assembly in diabetic conditions. We found that, in addition to matrix assembly, acetylation stimulated cell attachment to FN, integrin activity, and cell contractility. Integrin activation is regulated by interactions with the ECM and by inside-out signaling through cytoplasmic domain interactions with intracellular proteins [38,39]. The FERM domain-containing proteins talin and kindlin bind to adjacent regions of the β1 integrin cytoplasmic tail [34] and synergize to promote full integrin activation [40,41,42]. K794 is part of the kindlin binding motif in β1 integrin and we detected less kindlin-2 co-localization with β1(K794R) than with β1(K794Q) integrins suggesting that modifications at this site affect the composition of integrin-FN complexes. Others have shown that depletion of kindlin-2 in glomerular podocytes reduced integrin activation, cell adhesion, and FN matrix deposition [43].

Lysine acetylation can affect protein function by neutralizing lysine’s positive charge and by changing side chain structure, thereby altering protein–protein interactions, subcellular distribution, and stability [15]. Comparison of how kindlin and talin interact with their respective sites on the β1 integrin tail allows us to speculate about the potential effects of acetylation. Structural studies show that in β1D integrin binding to talin-2, the Ile side-chain of the β1 NPIY motif is buried in a hydrophobic pocket of talin that is formed by Ile and Leu residues (L356, I399, I402, L403) [44]. The comparable binding pocket in kindlin-2 maintains the hydrophobic nature (I654, F609 in equivalent positions to I399, L356 from talin), but is wider due to substitutions with smaller hydrophilic residues (S657, T658 instead of I402, L403). In the structure of kindlin-2 with a bound β1 integrin peptide, K794 in the β1 NPKY motif does not directly interact with kindlin [45]. This orientation is consistent with the observation that the K794A mutation had no negative effect on kindlin binding to the β1A tail in vitro [40]. However, if the positive charge of K794 were neutralized by acetylation, this would raise the possibility for hydrogen bonding with kindlin. It is tempting to speculate that the side chain of an acetylated K794 (or the smaller Gln in the K794Q mutant) could interact with this kindlin-2 pocket, augmenting the affinity of the NPAcKY motif for kindlin-2 and resulting in enhanced integrin activation and increased FN binding.

Integrin activity could be affected by the presence of cytoplasmic adapter proteins other than kindlin. For example, ICAP-1, a β1 integrin-specific adapter protein with a role in FN fibrillogenesis, negatively regulates kindlin-2 binding to β1 integrin [46]. The affinity of ICAP-1 for β1 integrin could hamper the recruitment of kindlin to the β1 tail. Modulating ICAP-1 affinity by post-translational modification of β1 would be an exciting possibility, and has been suggested for the recruitment of β1 integrins into podosomes by phosphorylation at S785 [47]. When analyzing the structure of the β1 integrin tail bound to ICAP-1, it is apparent that the side chain of K794 is solvent exposed and potentially available for modification [48]. Perhaps acetylation at K794 facilitates exchange of ICAP-1 for kindlin-2 to modulate integrin activity.

Along with integrin-bound talin and membrane PI4,5P2 lipids, kindlin-2 dimerization could be involved in inducing integrin clustering and full activation of integrins [45,49]. A global analysis of acetylation stoichiometry showed that most acetylation occurs at a very low stoichiometry [50]. This may not be a problem since clusters of three integrins are sufficient to activate FN signaling [51] and individual ligand spacing of 50 nm is able to induce integrin-dependent spreading [52]. Recent super-resolution imaging suggests that integrins are organized into nanoclusters composed of 50 to 100 receptors [53]. Perhaps only a few integrins per nanocluster are actually engaged in the adhesion and signaling process. Thus only a few acetylation events might be sufficient to stimulate integrin activity and subsequent pathways leading to FN assembly.

The increase in FN matrix assembly with expression of β1(K794Q) integrin was further enhanced by treatment with an HDAC inhibitor demonstrating that acetylation of other proteins is involved in assembly. Acetylome studies have identified acetylated proteins within interaction networks involved in cell adhesion and remodeling of the actin cytoskeleton such as talin-1 and vinculin [22,50]. Modification of a subset of proteins at low stoichiometry suggests that the effects of acetylation on different proteins might be combinatorial. We predict that acetylated adhesion/cytoskeletal proteins work together to promote FN matrix assembly.

A known consequence of HDAC inhibition is increased histone acetylation, which allows for changes in gene expression [54,55]. RNA-sequence analysis of mesangial cells treated with SBHA did not identify significant changes in overall gene expression (M.E.V., unpublished observations) and qPCR analyses of these cells showed no change in expression of FN or α5β1 integrin. Lysine acetylation could affect proteins by blocking ubiquitination or SUMOylation, thereby altering protein turnover. Interestingly, integrin β1 K794 has been identified as a target for SUMOylation [56]. However, mutation of K794 did not affect integrin levels since we did not detect differences in total integrin levels between β1(WT) and mutant β1 cells by immunoblotting or in surface integrin expression by flow cytometry. While histone acetylation and SUMOylation mechanisms are not directly related to FN or integrin functions, they may affect associated proteins that appear to be involved in regulating FN assembly.

Our work shows that increases in glucose levels affect ECM assembly through multiple mechanisms. We have previously shown that glucose metabolism stimulates FN matrix assembly by activating α5β1 integrin via protein kinase C in mesangial cells [4]. Mesangial cells also up-regulate their FN assembly when interacting with an ECM that is modified by glucose metabolites through non-enzymatic glycation [57]. Acetylation of mesangial cell β1 integrin represents a third mechanism for increasing ECM assembly in response to elevated glucose levels. SAHA and other HDAC inhibitors are currently approved or in clinical trials in the United States for treatment of various cancers. However, the adverse effect profiles of these inhibitors reduce their utility as treatments for diabetic kidney disease [58,59]. Our study indicates that they could have a deleterious effect by promoting fibrosis. Perhaps the best way to target the accumulation of ECM in kidney fibrosis is to directly control FN assembly and FN’s role as a foundational matrix for deposition of other ECM proteins.

## Figures and Tables

**Figure 1 cells-09-00655-f001:**
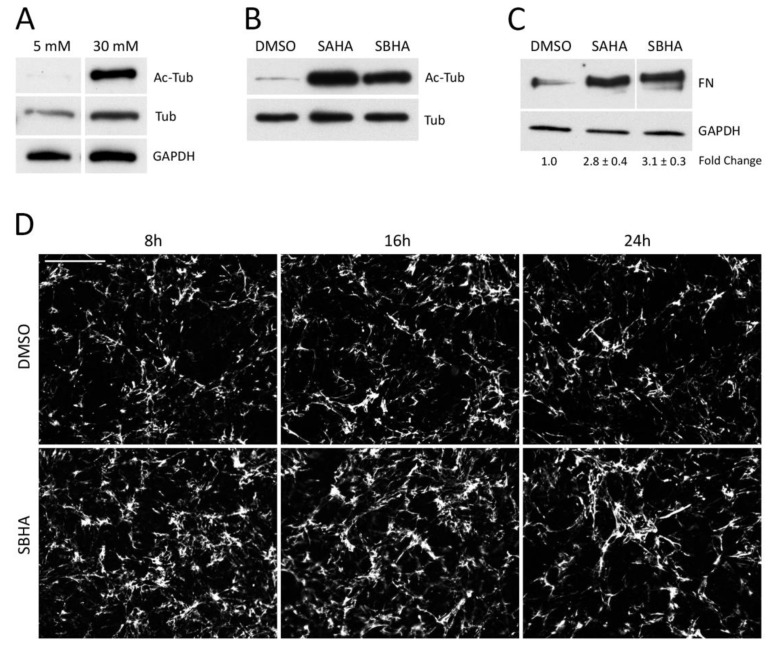
Increased acetylation promotes fibronectin (FN) matrix assembly. (**A**) Mesangial cells grown in 5 mM or 30 mM glucose for 48 h were lysed in RIPA buffer. (**B**,**C**) Mesangial cells grown in 30 mM glucose were treated with either 5 μM suberoylanilide hydroxamic acid (SAHA) or suberohydroxamic acid (SBHA) or DMSO (vehicle control) in medium containing 20 μg/mL human plasma FN for 24 h before lysis in deoxycholate (DOC) buffer. (**A**,**B**) RIPA and DOC-soluble lysates were immunoblotted with antibodies against acetyl-tubulin (Ac-Tub), tubulin (Tub) or GAPDH as indicated. (**C**) The DOC-insoluble fraction was immunoblotted with HFN7.1 anti-FN monoclonal antibody. Fold-changes are the average of three independent experiments (*p* < 0.01 for treatment compared to DMSO). Representative blots are shown and samples in each panel are from the same blot and exposure time. (**D**) Mesangial cells were grown as in (B,C) for 8 h, 16 h, and 24 h before staining with HFN7.1 antibody. The mean fluorescence intensity of 6 randomly selected fields per condition was measured using Image J software. Three independent experiments were quantified and average fold-changes were calculated of SBHA samples relative to DMSO samples. Mean ± SEM values are 1.36 ± 0.14 at 8 h, 1.20 ± 0.07 at 16 h, and 1.15 ± 0.05 at 24 h. Representative images are shown for each condition. Scale bar = 50 μm.

**Figure 2 cells-09-00655-f002:**
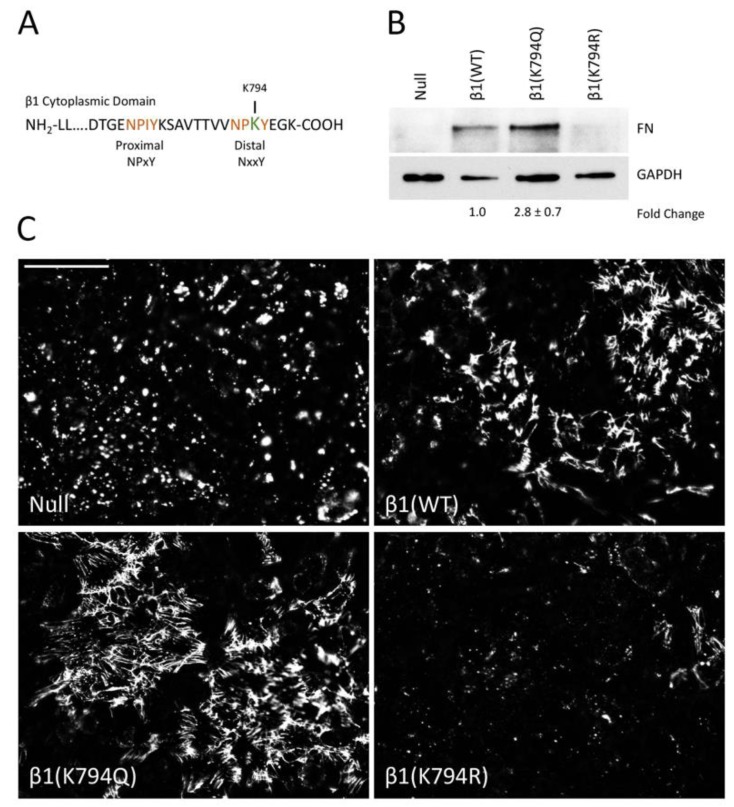
Integrin β1(K794Q) acetylated mimetic promotes FN matrix assembly. (**A**) The amino acid sequence of the mouse β1 integrin cytoplasmic tail shows the membrane-proximal and -distal NPxY motifs (orange) and K794 (green). (**B**) The DOC-insoluble fractions of GD25 β1(null), β1(WT), β1(K794Q), and β1(K794R) cells were immunoblotted with anti-FN antiserum (R184). The corresponding DOC-soluble fraction was immunoblotted with anti-GAPDH antibodies as a loading control. Fold-change is the average of four independent experiments (*p* < 0.03). A representative blot is shown. (**C**) Representative images of immunofluorescence staining of FN matrix from GD25 cell lines are shown. Fluorescence intensities were measured and calculated average fold-changes from three experiments compared to β1(WT) cells are 1.30 ± 0.01 for β1(K794Q) and 0.46 ± 0.03 for β1(K794R). Scale bar = 50 μm.

**Figure 3 cells-09-00655-f003:**
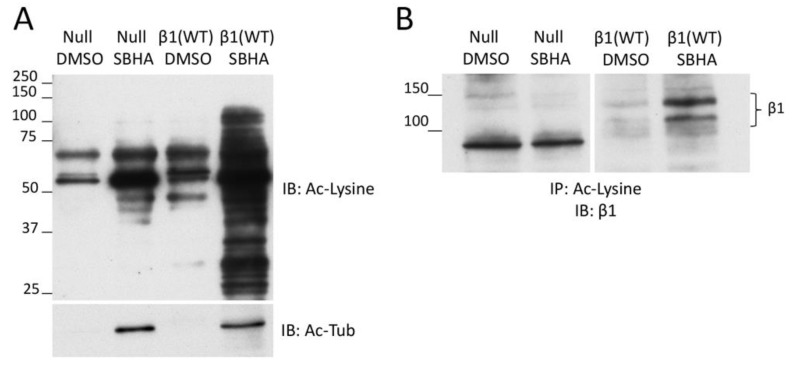
Deacetylase inhibitor treatment promotes β1 integrin acetylation. (**A**) GD25 β1(null) and β1(WT) cells treated with 5 μM SBHA or DMSO were lysed and equal total protein amounts were immunoblotted with anti-acetyl-lysine (Ac-Lysine, top) and anti-acetyl-tubulin (Ac-Tub, bottom) antibodies. Acetyl-tubulin is the most prominent band in the Ac-lysine blot, which was over-exposed to illustrate the presence of other acetylated proteins. (**B**) Anti-acetyl-lysine conjugated beads were incubated with cell lysates containing 1 mg total protein per condition for 2 h at 4 °C. Immunoprecipitated proteins were eluted in SDS buffer and immunoblotted for β1 integrin. Calculated molecular weights of β1 bands in panel B and Appendix A are the same.

**Figure 4 cells-09-00655-f004:**
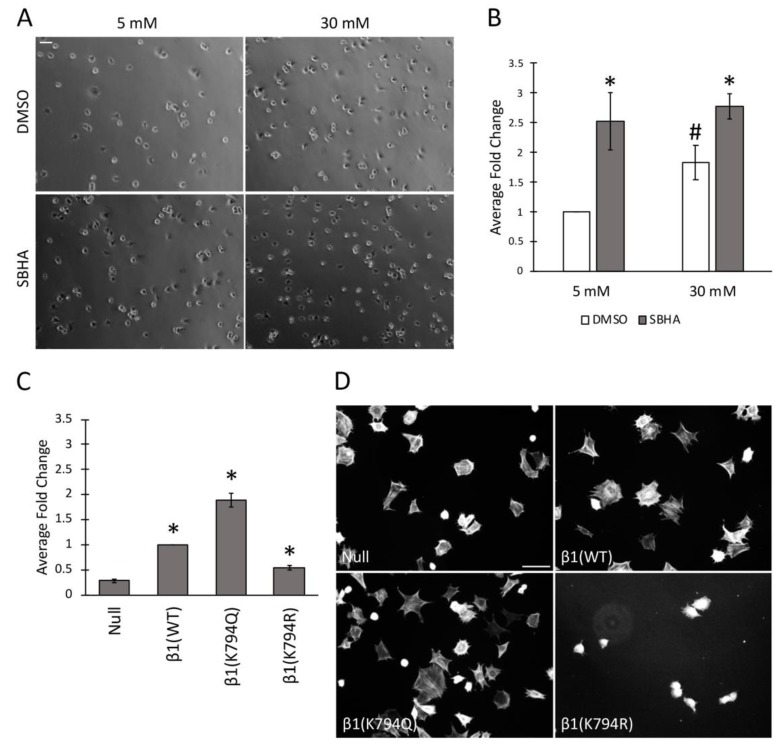
Acetylation promotes cell attachment to FN and alters cell spreading. (**A**–**C**) Mesangial cells conditioned in 5 mM or 30 mM glucose and GD25 cells were treated with 5 μM SBHA or DMSO and then allowed to attach to FN-coated surfaces for 30 min before fixation and cell counting. (**A**) Phase images of attached mesangial cells are representative of 3 independent experiments. Scale bar = 50 μm. (**B**,**C**) Attached cells were counted in 6 fields of view per condition and three experiments were averaged. Average fold change in number of attached cells was calculated and graphed relative to mesangial cells in 5 mM glucose/DMSO (**B**) or β1(WT) cells (**C**). * *p* < 0.05 compared to β1(null) (**C**) or compared to DMSO (**B**); # *p* < 0.05 compared to 5 mM (**B**). (**D**) GD25 cells were seeded on a FN coat in DMEM without serum for 4 h before staining with phalloidin. Images are representative from three independent experiments. Scale bar = 50 μm.

**Figure 5 cells-09-00655-f005:**
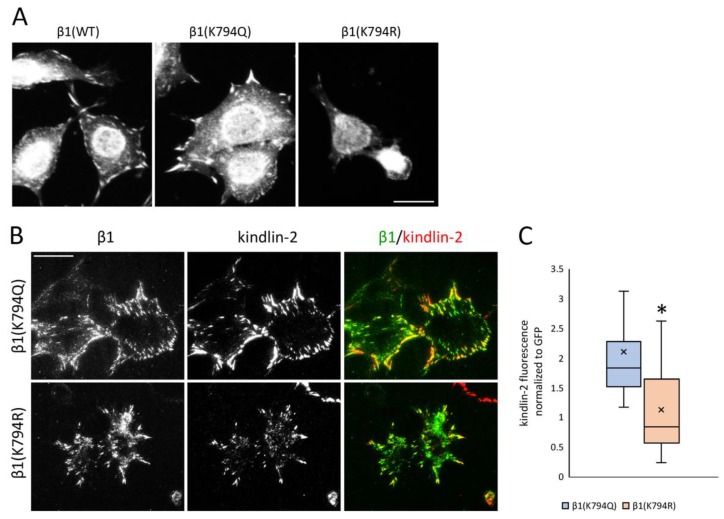
Kindlin-2 recruitment to β1 integrin focal adhesions. (**A**) Individual GD25 cells are shown. Cells were allowed to spread on a FN-coated surface for 4 h followed by fixing and staining with anti-kindlin-2 antibodies and visualization by epifluorescence. Scale bar = 10 μm. Fields of kindlin-2 stained cells are shown in Appendix A. (**B**) TIRF microscopy was used to visualize immunostained kindlin-2 and GFP-tagged β1 integrin in β1(K794Q) and β1(K794R) cells on a FN-coated surface. Scale bar = 10 μm. (**C**) Average fluorescence intensities of kindlin-2 normalized to GFP for n = 30 cells per condition. Box-plot crossline is the median and x indicates the mean. * *p* < 0.001.

**Figure 6 cells-09-00655-f006:**
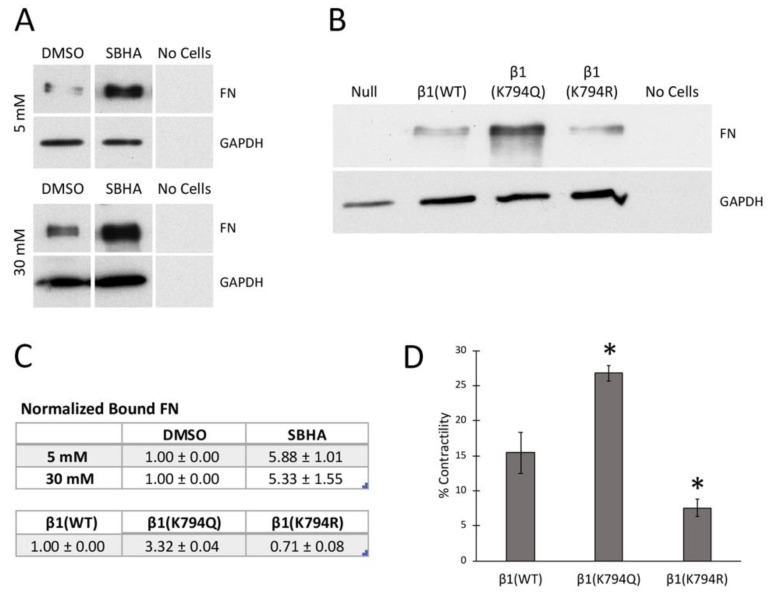
Acetylation promotes integrin activation. (**A**) Mesangial cells grown in 5 mM (top) or 30 mM (bottom) glucose and SBHA or DMSO and (**B**) GD25 cells expressing various integrins were incubated with FN in suspension. Cell-bound FN was detected by immunoblotting with HFN7.1 anti-human FN antibody and GAPDH (as loading control). Representative blots are shown, and FN lanes in 5 and 30 mM panels are from the same blot and exposure time. (**C**) Average fold-changes in bound FN (± SEM) were calculated from three independent experiments for the conditions in (**A**,**B**). *p* < 0.02 for each condition compared to control. (**D**) Changes in matrix area with contraction by the indicated cells within a fibrin-FN matrix were averaged for three independent experiments and data are expressed as the mean ± SEM. * *p* < 0.05 relative to β1(WT) area.

**Figure 7 cells-09-00655-f007:**
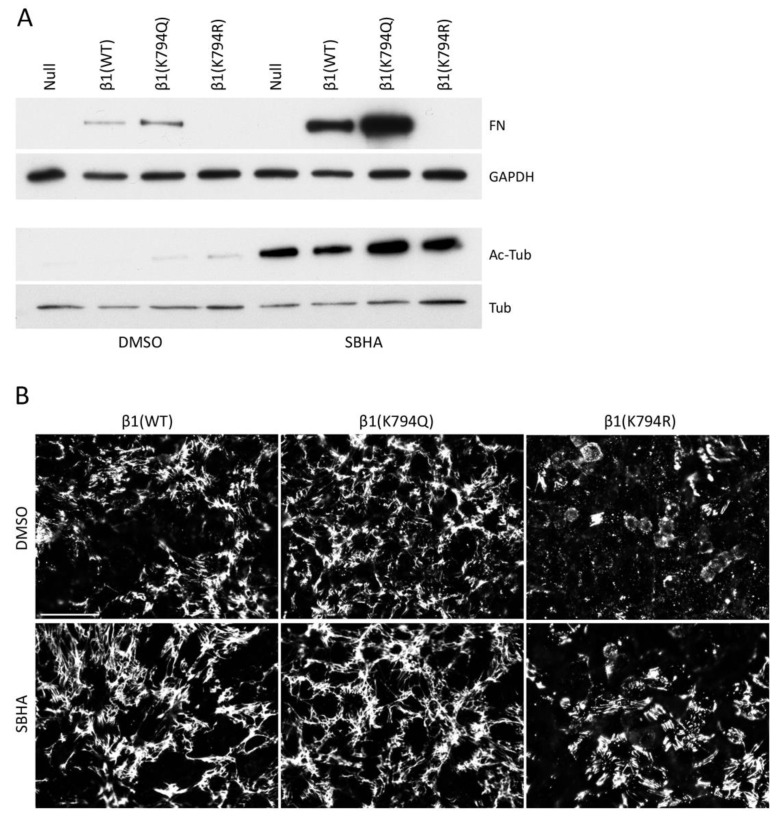
Lysine acetylation of additional proteins is necessary for maximum FN matrix assembly. GD25 cells were treated with 5 μM SBHA or DMSO for 24 h before DOC lysis and immunoblotting of DOC-insoluble FN (**A**) and immunofluorescence staining with anti-FN antiserum R184 for visualization of FN matrix (**B**). (**A**) DOC-soluble fractions were also immunoblotted with anti-GAPDH, anti-acetyl-tubulin, and anti-tubulin antibodies. Fold-changes in fluorescence intensities between DMSO and SBHA-treated cells were calculated and normalized to β1(WT) DMSO cells (set to 1). Values are 1.31 ± 0.01 (β1(K794Q) DMSO), 0.43 ± 0.01 (β1(K794R) DMSO), 1.26 ± 0.13 (β1(WT) SBHA), 1.61 ± 0.20 (β1(K794Q) SBHA), and 0.74 ± 0.15 (β1(K794R) SBHA). Representative images are shown for each condition. Scale bar = 50 μm.

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
