# Peer review of "Stimulation of Fibronectin Matrix Assembly by Lysine Acetylation"

_cells, 2020, doi:10.3390/cells9030655_

Round 1

Reviewer 1 Report

The manuscript by Vega and colleagues is the resubmission of the previous paper #cells-651283 entitled “Elevated glucose levels stimulate fibronectin matrix assembly via acetylation of the integrin beta-1 tail”. The manuscript remained substantially unchanged and I’m sorry to confirm my judgment. The make-up operation does not avoid doubts about the critical points reported in the first review.

In particular:

1) A main message of the manuscript is that high glucose induces β1 integrin acetylation on lysine 794, and this has not been demonstrated. 

2) Some important controls are mentioned as “not shown”, why don’t the authors show them (e.g. total FN in figure  1C and 2B)?

3) Patched western blots are not acceptable, especially if quantitative WB.

4) Figure 5 and S4 remain unconvincing about the difference in Kindlin-2 localization in focal adhesions. Moreover, the data interpretation about Kindlin-2 remain obscure to me. The authors stated that: “Interestingly, in β1(K794R) cells, the kindlin-2 signal was fainter and was not always present in integrin-positive focal adhesions. Kindlin-2 levels were equivalent in all β1 integrin-transfected cells (Figure S4C).” There is less Kindlin-2 (fainter signal also quantified in panel 5C and less kindlin-2 in β1(K794R) cells shown in Fig. S4C) or Kindlin-2 levels were equivalent? Moreover, in GD25 cells without β1 integrin, Kindlin is missing (Fig. S4C). This point is not discussed: is β1 integrin (and Lysine 794) essential for Kindlin-2 stabilization/expression?

Reviewer 2 Report

MAIN COMMENTS

The main weakness remains the same in the revised manuscript as previously, i.e. no data is presented showing that acetylation of b1 integrin is linked to the observed effects on cell-fibronectin interactions. The authors have added a section in the Discussion that well explains their thinking, but as they also write, it is a speculative model. I actually like the idea that removal of the positive charge in the kindlin binding site by acetylation of K794 could be a physiological or pathophysiological reaction affecting b1 integrin functions, and I think that it can be published as a hypothetical model. However, then it is important to rewrite the text accordingly, and not to give the reader another impression. Examples of sentences that need to be altered include:

Lines 16-18: “The matrix effects were linked to acetylation of lysine 794 (K794) in the b1 integrin cytoplasmic domain using cells expressing acetylated (K794Q) and non-acetylated (K794R) mimetics.”

Possible variant: The matrix effects are suggested to be linked to acetylation of lysine 794 (K794) in the b1 integrin cytoplasmic domain based on studies of cells expressing acetylated (K794Q) and non-acetylated (K794R) mimetics.

Lines 19-20: “We show that modification at K794 affects FN assembly”

Possible variant: We show that mutations at K794 affects FN assembly

Lines 64-65: “ … identify a role for b1 integrin modification in cell-FN interactions”

Possible variant: … identify a possible role for b1 integrin modification in cell-FN interactions

Lines 325-326: “ … cells expressing the fully acetylated versus non-acetylated mutant integrins suggest”

Possible variant: … cells expressing the mutant integrins suggest

Lines 338-339: “These attachment data from two distinct cell types link b1 integrin acetylation state with the ability of cells to attach to a FN-coated substrate.

Possible variant: These attachment data from two distinct cell types supports the possibility that b1 integrin acetylation state affects the ability of cells to attach to a FN-coated substrate.

Lines 406-407: “ … we show that K794 modification promotes integrin activation in order to enhance FN matrix assembly.

Possible variant: … we show that K794Q mutation promotes integrin activation in order to enhance FN matrix assembly.

Lines 421-422: “To determine if acetylation of other proteins in addition to b1 integrin contributes to matrix assembly”

Possible variant: To determine if acetylation of other proteins than b1 integrin contributes to matrix assembly

Lines 429-430: “… acetylation of lysines in addition to the K794 site further stimulates matrix assembly.”

Possible variant: … acetylation of lysines other than the K794 site further stimulates matrix assembly.

Lines 453-455: “… the inability to acetylate b1 in cells expressing the 1(K794R) mutant critically hampered assembly of FN matrix”

Possible variant: … the inability to acetylate b1 in cells expressing the 1(K794R) mutant may explain the critically hampered assembly of FN matrix

The authors speculate that AcK794 and K794Q increase b1 integrin activation and thereby increased fibronectin binding and polymerization (lines 492-493), but the new fig S2D shows that this is not the case (mAb 9EG7 data). Please clarify.

The authors speculate about the role of AcK794 in regulating affinities with kindlin-2 and ICAP-1. This could be tested directly in interaction studies using synthesized b1-peptides having K794 or acetylated K794. The authors write that maybe the affinities may be too small to detect, but that we don´t know without testing.

I do not agree with the answer to my previous Minor comment 4. In contrast to the author´s statement, it is well known that different integrin beta-subunits compete for associating with limiting amounts of aV in the ER. b3 is somewhat preferred over b1 by aV, but increasing the expression of b1 will result in more aVb1 and less aVb3. Since aVb3 will contribute the adhesion to fibronectin, the expression of a b1 mutant which is partially defect would be expected to result in the observed effects on cell spreading (i.e. GD25b1K794R spreading slower than GD25 null). Thus, it would be highly relevant to measure the surface levels of b3 in GD25 null cells and GD25 transfected with the b1 variants.

A related issue is that the presence of aVb3 will complicate all interpretations of the effects of b1 mutations on fibronectin interactions. More clear data would be obtained if the contribution of aVb3 was blocked, which can easily be done by the use of aVb3-selective cyclic RGD-peptides.

OTHER QUESTIONS OF INTEREST:

Why are there much more lysine-acetylated proteins in GD25b1 than in b1-deficient GD25 cells (fig 3A)?

What is the lower strong band in the beta1 WB of the lysin-acetylated proteins in fig 3B (lane 4)? The upper band is said to co-migrate with the band(s) in fig S2A, but also the lower band is marked as b1.

Why is there almost no kindlin-2 in beta1 integrin null GD25 cells (fig S4C)? This interesting and unexpected finding is not commented on in the manuscript.

NOTE

Supplementary fig S1C is referred to as fig S1B in the text (line 234).

Round 2

Reviewer 1 Report

If it was not authors’ goal to show that beta1 integrin is acetylated on lysine 794 by high glucose conditions, the manuscript  is a mix of  different observations not completely characterized. But this assessment is the responsibility of the editor.

Regarding the “not shown” controls of total FN, authors stated at line 236: “Quantitative PCR analysis of FN mRNA levels showed that neither glucose concentration nor HDAC inhibitors affected FN expression (data not shown)”. I am quite surprise that the authors prefer to object to this point instead of showing the data obtained on endogenous FN.

As for the quality of WB panels, I don’t think it is difficult to show a well-loaded WB. However, if for the Editor the data presentation meets the journal standards, I do not insist.

Describing figure 2B, it is stated that “Cells expressing the acetyl-mimic b1(K794Q) integrin had the highest amount of FN matrix compared to the b1(WT) cells 2.8 +/- 0.7-fold)”. Quantitative analysis of the bands reveals that this is true only if normalization to GAPDH (as described in MM lines 147-8) is not done, otherwise the difference is not appreciable. How do the authors justify that?

Author Response

Please see attached letter to Editor Dr. Smith. 

Reviewer 2 Report

I am satisfied with the response to all my points by the authors. I also wish them good luck in their future research. 

Author Response

No comments needed to be addressed.

This manuscript is a resubmission of an earlier submission. The following is a list of the peer review reports and author responses from that submission.

Round 1

Reviewer 1 Report

This manuscript reports a potential interesting connection between glucose levels and fibronectin (FN) ECM assembly drives by acetylation of integrin β1 tail on lysine 794. The authors present a series of experiments demonstrating that protein acetylation enhances FN matrix assembly and high glucose levels increase attachment on FN of mesangial cells, and they demonstrate that Integrin β1(K794Q) acetylated mimetic produces the same effects in fibroblasts.

The manuscript is carefully and clearly written, and it is potentially interesting, but some issues have to be addressed before publication.

Major issues.

The major criticism is that the article does not really demonstrate a clear connection between glucose levels and FN assembly driven by integrin β1acetylation, because the hypothetical effect of high glucose levels is obtained indirectly by deacetylase inhibitors. A key experiment is missing: the demonstration that integrin β1 acetylation increases, possibly on lysine 794, in high glucose condition in comparison with low glucose levels. Results (line 209) and Figure 1A. This panel should prove that “elevated glucose levels cause an increase in lysine acetylation” in mesangial cells. It is not clear how was the acetylated tubulin quantification done. Were the band intensities normalized for GAPDH or total tubulin? Anyway, it seems that high glucose induces the total amount of tubulin instead of the percentage of acetylated tubulin and the statement “Therefore, growth of mesangial cells in high glucose media stimulates lysine acetylation” is not really convincing, even considering that at 30 mM the amount of GAPDH is higher. In Figure 6A, where FN binding to mesangial cells is shown, apparently there is no difference between 5 mM and 30 mM glucose treated control cells (comparative quantification of these two points is missing). This is problematic for the paper. For me is difficult to accept fragmented western blot pictures (Fig. 1A and Fig. 6A), especially when quantification is applied. In Figure 5, the pictures do not convince me enough that Kindilin-2 recruitment is different between the K794 mutants. First, only one cell with double (β1/Kindilin-2) staining is shown  for β1(K794R) and second, it seems that less total Kindilin-2 is expressed in β1(K794R) cells instead of less localization of  Kindilin-2 at focal adhesion sites.

Minor issues.

Statistical analysis has to be done by ANOVA with a multiple comparison test instead of Student’s t-test. Materials and Methods line 73. Please specify the origine of the rabbit polyclonal anti-FN (R184) antiserum.

Reviewer 2 Report

In this manuscript Vega et al. concludes that lysine acetylation of integrin beta1 is necessary for FN matrix assembly, and that glucose-induced integrin acetylation contributes to matrix over-production in diabetic kidney disease. The conclusion is based on two approaches, i.e. the use of histone deacetylase inhibitors (SAHA and SBHA) and on mutation of lysine 794 in the cytoplasmic domain of integrin beta1 to glutamine and arginine to mimic acetylated and non-acetylated beta1, respectively.

Addition of the histone deacetylase inhibitors to cultured mesangial cells increased the amount of acetylation on lysine 794 in the cytoplasmic domain of integrin beta1 and also caused more fibronectin polymerization into the extracellular matrix by the cells. GD25 cells expressing beta1(K794Q) formed more fibronectin matrix than GD25 cells expressing wildtype beta1, and the beta1(K794R) cells failed to polymerize fibronectin. Cell adhesion, spreading, and gel contraction were enhanced in beta1(K794Q) cells and reduced in beta1(K794R) compared to wt beta1 cells. These effects were likely linked to the ability of the integrin to bind kindlin2; beta1(K794Q) efficiently bound kindlin2 while beta1(K794R) bound poorly.

The results presented are clear and well presented. However, a main problem with the present study is that glutamine is simply not a reliable mimetic for acetylated lysine although it has been used for that purpose in several articles. The second problem is that histone deacetylase inhibitors affect many proteins and cannot by themselves link integrin acetylation to fibronectin polymerization.

Major comment

1. The main conclusion that acetylation of lysine 794 integrin beta1 is necessary for FN matrix assembly (Abstract line 22, Results line 407,) is not compatible with previous reports on how beta1 integrins (i.e. a5b1) promote fibronectin polymerization and how they bind to kindlin2:

1a) EHJ Danen et al (J Cell Biol 2002, 159: 1071–1086) demonstrated that a chimeric beta1/beta3 protein, where the extracellular and transmembrane domains of beta1 are fused with the cytoplasmic beta3 domain, is equally efficient in promoting fibronectin polymerization as wt beta1 although beta3 has a threonine in the position of lysine794 in beta1 (the NPKY motif in beta1 is NITY in beta3).

1b) Kindlin2 was shown to bind to both beta1 and beta3 cytoplasmic domains via the NxxY motif and the adjacent T/ST. Replacement of K794 with alanine in beta1 did not affect the binding of kindlin2 (while replacements of either T788, T789 or Y795 with alanine abolished the binding) (DS Harburger et al., J Biol Chem. 2009, 284:11485-11497). Furthermore, the crystal structures of kindlin2 in complex with the C-terminal part of the beta1 and beta3 cytoplasmic domains were found to be almost superimposable; the NPKY and NITY are positioned in the same way in the binding grove of kindlin2 (H Li et al., Proc Natl Acad Sci U S A 2017, 114: 9349-9354).  

1c) As a general comment, the structural similarity between the side chains of glutamine and acetyl lysine is limited, e.g. the sizes differ markedly (4 vs 7 atoms in length). Replacement of lysine with glutamine may be tolerated by binding partners, in this case kindlin2, because of its small size (as threonine and alanine in the above examples) rather than by some resemblance to an acetylated lysine. Similarly, changing lysine to the larger arginine may prevent the interaction because it does not fit into the binding site in kindlin2 rather than being caused by the absence of acetylation.

1d) Thus, while acetylation of lysine 794 clearly is not required for the studied functions of beta1 integrins in contrast to the statments in the manuscript, the presented data may indicate that this modification enhances the interaction with kindlin2 or other proteins. One possible way to obtain support for that conclusion could be to use the available data from H Li et al., Proc Natl Acad Sci U S A 2017, 114: 9349-9354 to make computer modelling of kindlin2 in complex with the beta1 NPxY motif where K794 is altered to Q, acetyl-K, and R. A revised version of the manuscript would also have to discuss the information in the articles referred to under 1b) and 1c).

Minor comments

2. The FACS analysis of levels of beta1 should be shown for the GD25 cells transfected with beta1 mutations, as well as the subunit a5 to ensure that the surface level of a5b1 is comparable between the different cell lines.

3. It would be relevant to analyze the conformation status of beta1(K794Q) and beta1(K794R) compared to wt beta1 by FACS using conformation-specific antibodies, e.g. 9EG7.

4. Fig 4D indicates that the beta1(K794R)-expressing cells spread less the beta1 null cells, suggesting a negative effect on aVb3-mediated spreading by beta1(K794R). An obvious possibility is that the expressed beta1 competes with beta3 for association with aV and thereby affects the amount of aVb3 formed. To clarify this issue, also the surface level of beta3 should be analyzed.